# Foundation Models for Enhanced Exploration in Reinforcement Learning

## Abstract

Reinforcement learning agents often struggle with sample inefficiency, requiring extensive interactions with the environment to develop effective policies. This inefficiency is partly due to the challenge of balancing exploration and exploitation without the abstract reasoning and prior knowledge that humans use to quickly identify rewarding actions. Recent advancements in foundation models, such as large language models (LLMs) and vision-language models (VLMs), have shown human-level reasoning capabilities in some domains but have been underutilized in directly selecting low-level actions for exploration in reinforcement learning. In this paper, we investigate the potential of foundation models to enhance exploration in reinforcement learning tasks. We conduct an in-depth analysis of their exploration behaviour in multi-armed bandit problems and Gridworld environments, comparing their performance against traditional exploration strategies and reinforcement learning agents. Our empirical results suggest foundation models can significantly improve exploration efficiency by leveraging their reasoning abilities to infer optimal actions. Building on these findings, we introduce Foundation Model Exploration (FME), a novel exploration scheme that integrates foundation models into the reinforcement learning framework for intelligent exploration behaviour. We use VLMs and demonstrate that they can infer environment dynamics and objectives from raw image observations. This means FME only requires the action space as environment-specific manual text input. We find that agents equipped with FME achieve superior performance in sparse reward Gridworld environments and scale to more complex tasks like Atari games. Moreover, the effectiveness of FME increases with the capacity of the VLM used, indicating that future advancements in foundation models will further enhance such exploration strategies.

## 1 Introduction

Reinforcement learning enables agents to learn optimal policies through interactions with an environment by observing states, taking actions, and receiving rewards. The goal is to find a policy that maximizes the expected cumulative rewards. Despite its broad applicability in fields like healthcare, robotics, logistics, finance, and advertising, reinforcement learning often suffers from sample inefficiency, requiring vast amounts of data and numerous trial-and-error iterations to develop effective policies. A crucial factor of being sample efficient is balancing exploration, the act of seeking new knowledge about potential rewards, with exploitation, the use of known rewarding actions.

Scalable exploration strategies in reinforcement learning typically introduce random actions or rely on heuristics. While these methods are general and widely applicable, they can incur redundant environment interactions, especially in cases where the path to rewards would be apparent to a human observer. For instance, in environments where visual cues clearly indicate the goal, such as navigating a maze with visible exits, traditional methods may waste time exploring irrelevant paths. This inefficiency stems from the absence of abstract reasoning and prior knowledge in conventional reinforcement learning algorithms, limiting their effectiveness in complex environments.

Recent advances in foundation models, particularly large language models (LLMs) and vision-language models (VLMs) have demonstrated remarkable capabilities in retaining vast amounts of knowledge and exhibiting human-level reasoning when trained on massive datasets of text and images (Brown et al., 2020; Team et al., 2023; Achiam et al., 2023; Touvron et al., 2023; Jiang et al.,

2024; Team et al., 2024). Prior works have explored leveraging foundation models in reinforcement learning, often focusing on text-based environments or using them as auxiliary components for tasks like reward shaping and goal generation (Klissarov et al., 2024; Du et al., 2023). When used for decision-making, foundation models are frequently equipped with hand-crafted abstract skills to enable autonomous action (Wang et al., 2024b). However, these approaches often involve significant prompt engineering and impose constraints that limit the agents' capabilities and generalization potential through reliance on text-based interfaces or predefined skill sets. The lack of involvement of foundation models in low-level decision-making means that their fundamental capacity to address traditional reinforcement learning exploration challenges is currently unexplored.

In this paper, we aim to fill this gap by conducting a fundamental study of the potential of foundation models in traditional reinforcement learning settings. We analyse foundation models' exploration behaviour in multi-armed bandit settings and compare their performance to traditional exploration strategies. Our findings reveal that these models can outperform well-established exploration algorithms in certain settings. We extend this analysis to Gridworld environments, where we observe efficient navigation behaviour of foundation models in fully observable deterministic settings but suboptimal performance when stochasticity and partial observability are introduced. These experiments highlight the complementary strengths and weaknesses of reinforcement learning algorithms and foundation models in decision-making, underlining the need for a hybrid approach.

To this end, we propose FME (Foundation Model Exploration), a multi-step adaptable exploration scheme that integrates foundation models with reinforcement learning agents by letting the foundation model control the agent's behavior for certain periods of time. We demonstrate that, by using VLMs, we can equip an arbitrary reinforcement learning agent with FME without extensive prompt engineering or having to provide explicit environment descriptions. Simply by specifying the action space in natural language, the VLM can use its prior knowledge and reasoning to understand the environment's dynamics and objectives from visual inputs, enabling it to produce effective sequences of actions that assist the agent in accumulating rewards.

We evaluate and analyze FME in various environments, including sparse-reward Gridworlds and complex domains like Atari games. Our experiments demonstrate that FME can help agents explore significantly more efficiently in accumulating rewards. We also find that the performance gains increase as larger foundation models are used, suggesting that the approach benefits from the increased capacity. Lastly, using FME, researchers can provide manual priors in natural language to the agent, such as hints of achieving the environment's objective.

Our contributions can be summarized as follows:

- We conduct the first fundamental study of the capacity of foundation models in traditional reinforcement learning exploration challenges, providing insights into their strengths and limitations in such settings.
- We propose FME, the first method to utilize VLMs for enhancing the exploration efficiency of reinforcement learning agents through a temporally extended and adaptable exploration scheme that requires minimal prompt engineering.
- We empirically demonstrate the effectiveness of FME in improving exploration efficiency in Minigrid and Atari games.

The remainder of this paper is structured as follows. Section 2 reviews previous works. Section 3 performs an in-depth analysis of the potential of foundation models in the exploration aspect of reinforcement learning problems. Section 4 introduces FME and describes the corresponding technical details and experiments. Finally, Section 5 summarizes our findings and lays out the implications for future research.

## 2 RELATED WORKS

**Foundation models as auxiliary components.** In the context of reinforcement learning, LLMs have been used to shape the behaviour of the agents by reward shaping and goal generation. Klissarov et al. (2024) construct an intrinsic reward model that encodes the preference of an LLM between two observations in a text-based environment. Ma et al. (2024a) use foundation models to write high- and low-level code for robotic control movements to help with exploration in continuous control

domains. Triantafyllidis et al. (2023) use LLMs to rate actions in robotic manipulation tasks and use these ratings as intrinsic rewards to enhance exploration. If the underlying code of the environment is available, Ma et al. (2024b) demonstrates that an LLM can directly program a reward function for a given task description. Du et al. (2023) utilize an LLM to generate plausible goals for guiding the training of a goal-conditioned reinforcement learning agent. Similarly, Yang et al. (2023) train a multi-modal reinforcement learning agent that learns to imitate an expert LLM policy defined via PDDL. Rocamonde et al. (2024) use a VLM as zero-shot reward models for robotic control environments. Zhang et al. (2024) use LLMs for task-selection in open-ended learning settings. Finally, Xie et al. (2024) prompts LLMs to generate reward codes that are used to shape rewards for continuous control reinforcement learning agents. However, these approaches primarily leverage foundation models as auxiliary tools to influence agent behaviour indirectly through reward shaping or goal generation, rather than integrating them directly into the agent's action selection process. In contrast, our work directly incorporates foundation models into the agent's decision-making process, allowing them to select low-level actions to enhance exploration efficiency.

**Foundation models for decision-making.**    The reasoning abilities of large language models (LLMs) have been leveraged to produce high-level plans, which are executed through low-level skills to interact with the environment (Wang et al., 2024b; Lin et al., 2023). These low-level skills can be manually implemented behaviours (Zhu et al., 2023; Wu et al., 2023; Huang et al., 2022a), pre-trained machine learning models (Wang et al., 2023; Huang et al., 2022b; Song et al., 2023), or accessed via programming APIs (Wang et al., 2024a; Liang et al., 2022). While utilizing these skills reduces the computational cost of querying LLMs, the constraints they introduce can limit the agent's capabilities and generalization (Song et al., 2023). While these methods demonstrate the potential of foundation models in decision-making, they often rely on hand-crafted low-level skills or predefined behaviours to interact with the environment. In contrast, our approach differs by introducing a hybrid approach that eliminates the need for predefined skills or extensive prompt engineering by directly utilizing VLMs to infer environment dynamics and objectives from raw visual inputs.

## 3 EXPLORATION STUDY

In this section, we perform an in-depth analysis of the exploration behaviour of foundation models in traditional reinforcement learning settings. Our goal is to understand their fundamental capacity for handling the exploration-exploitation trade-off and to identify their strengths and limitations compared to traditional reinforcement learning algorithms. To achieve this, we choose to study two well-established environments: multi-armed bandit problems and Gridworld environments. The multi-armed bandit problem (used in Section 3.1) represents one of the most fundamental and widely studied paradigms for evaluating exploration strategies in reinforcement learning. This framework provides a simplified setting that focuses solely on the exploration-exploitation trade-off, making it ideal for analyzing the exploratory decision-making capabilities of foundation models in a controlled environment. Gridworld environments (used in Section 3.2), on the other hand, introduce spatial navigation and state transitions, offering a more complex setting to examine the decision-making and exploration abilities of foundation models.

### 3.1 MULTI-ARMED BANDITS

Our study first introduces the application of foundation models, particularly large language models (LLMs), to the domain of bandit exploration to study their fundamental capacity for handling the exploration-exploitation trade-off.

**Algorithms.**    The LLMs used for these experiments include GPT-3.5 and GPT-4 (Achiam et al., 2023), and Gemini 1.0 and Gemini 1.5 (Team et al., 2023). This diverse collection enables us to assess the decision-making abilities of LLMs across multiple model types and generations. Although the exact parameter counts for the GPT and Gemini models are unknown, performance benchmarks suggest that Gemini 1.0 aligns closely with GPT-3.5, and Gemini 1.5 with GPT-4 in terms of parameter range (Chiang et al., 2024). Each of these LLMs is prompted with the template as shown in Listing 1, which includes a raw memory of all previous arms pulled and corresponding observed rewards. The temperature parameter of the LLMs is set to 0. We compare the performance of the foundation

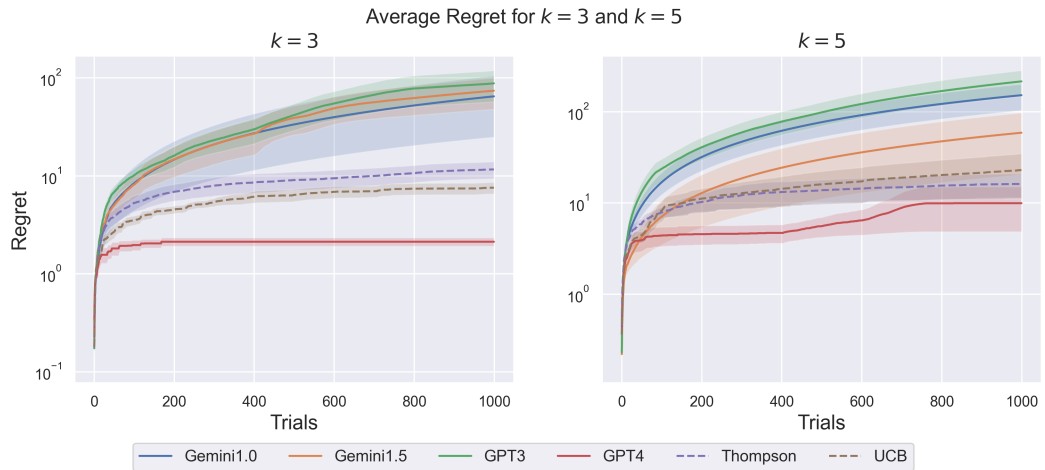

Figure 1: Averaged regret of the Bandit experiments of $k = 3$ and $k = 5$ for various LLMs, Thompson Sampling, and UCB.

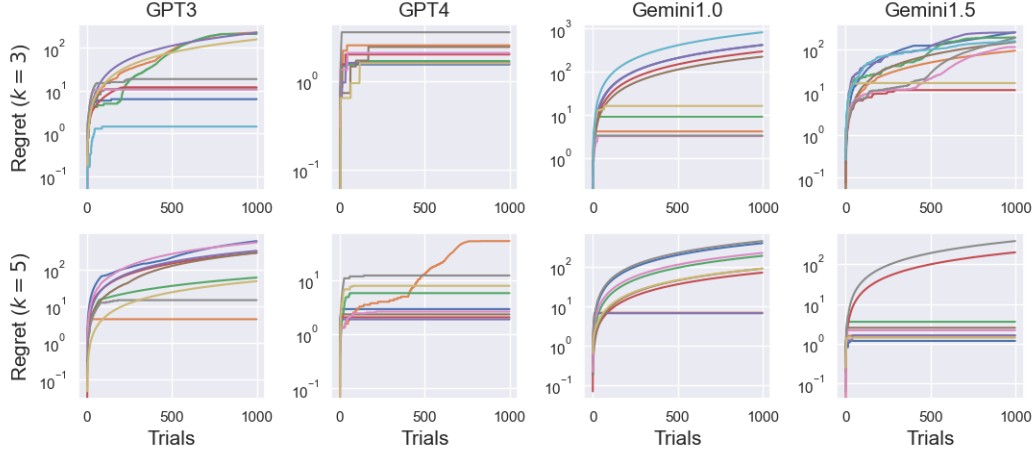

Figure 2: Individual seeds for each of the LLMs in the $k = 3$ (top) and $k = 5$ (bottom) settings, providing insights on the different strategies employed by each model.

models to two well-known efficient methods for bandit problems, Thompson Sampling (Thompson, 1933) and Upper-Confidence Bound (UCB) (Auer, 2002).

**Average regret analysis.** We first consider two $k$-armed bandit settings where $k = 3$ and $k = 5$. We consider Bernoulli arms where the probability of success $p_i$ of a given arm $i$ is uniformly sampled, i.e. $p_i \sim U(0, 1)$. Each approach is evaluated across ten seeds and performs 1000 trials. The averaged results can be found in Figure 1. For both $k = 3$ and $k = 5$ settings, we observe that most LLMs perform significantly worse than UCB and Thompson sampling. However, we observe that GPT-4 significantly outperforms all other methods in both settings, achieving a consistent lower regret compared to Thompson Sampling and UCB.

When looking at the individual seeds of each of the LLM models in visualized Figure 2, we observe that GPT-3 and Gemini 1.0 seem to commit to a single arm early on in most cases even though it is often not the optimal arm in. Gemini 1.5 seems to exhibit suboptimal exploratory behavior in the $k = 3$ setting, but interestingly performs relatively well in the $k = 5$ setting. Finally, we see that

GPT-4 generally performs an initial exploration phase and then commits to a single arm, which in nineteen out of twenty cases was indeed the optimal arm, reminiscent of the 'explore-then-commit' algorithm (Perchet et al., 2015). Furthermore, interestingly, in the remaining case, halfway through, it realizes that the arm it committed to may not be the optimal one, and it then continues to explore and eventually successfully find the optimal arm.

**Suboptimality gap analysis.** To investigate the surprising performance of GPT-4 further, we study its decision-making under different suboptimality gaps. In a 2-armed Bernoulli bandit setting, the suboptimality gap is defined as $\Delta := \max(p_0, p_1) - \min(p_0, p_1)$. That is, the suboptimality gap measures how much worse the suboptimal arm is when compared to the optimal arm. We consider 2-armed bandit settings where $\Delta \in \{0, 0.2, 0.4, 0.6\}$. This range allows us to evaluate the algorithms under different levels of difficulty:

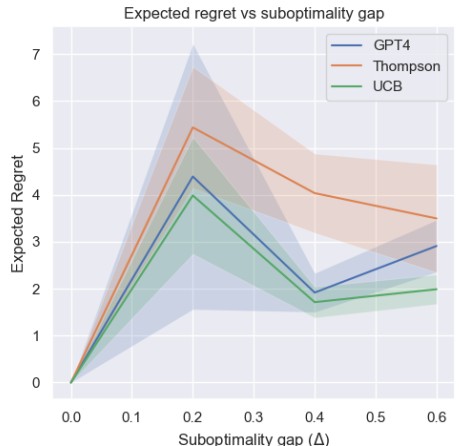

- **Moderate suboptimality gap** ($\Delta \in \{0.4, 0.6\}$): The optimal arm is much better than the suboptimal arm, making it easier for an algorithm to identify and exploit the optimal action quickly.

- **Small suboptimality gap** ($\Delta = 0.2$): This represents the most challenging scenario, where the difference between arms is subtle. Algorithms must carefully balance exploration and exploitation to identify the optimal arm without incurring excessive regret.

Figure 3: Suboptimality gap experiments for 2-armed bandit settings, comparing GPT-4, UCB, and Thompson Sampling.

We compare the performance of GPT-4 to Thompson Sampling and UCB and take the average regret across five random seeds for 500 trials for each of the algorithms. As can be seen in Figure 3, the performance of GPT-4 seems to be comparable to Thompson Sampling and UCB - each of the algorithms performs well in the moderate suboptimality gap settings, whereas they accumulate more substantial regret in the small suboptimality gap setting.

## 3.2 GRIDWORLDS

To further understand foundation models' decision-making and exploration abilities, we continue to examine their behaviour in two reinforcement learning settings where state transitions do matter. We evaluate the LLMs in two Gridworld settings that provide different exploration challenges and compare their performance to traditional reinforcement learning agents.

Listing 1: The prompt template used for the bandit settings.

```
**Context**
You are presented with {k} arms.  Each arm has a different probability of
    success. Your goal is to maximize the total reward by finding out
   which arm has the highest probability of success.

**Memory**
{memory}

**Available Arms**
{actions}

**Task**
Choose an action from the given list of available arms.
```

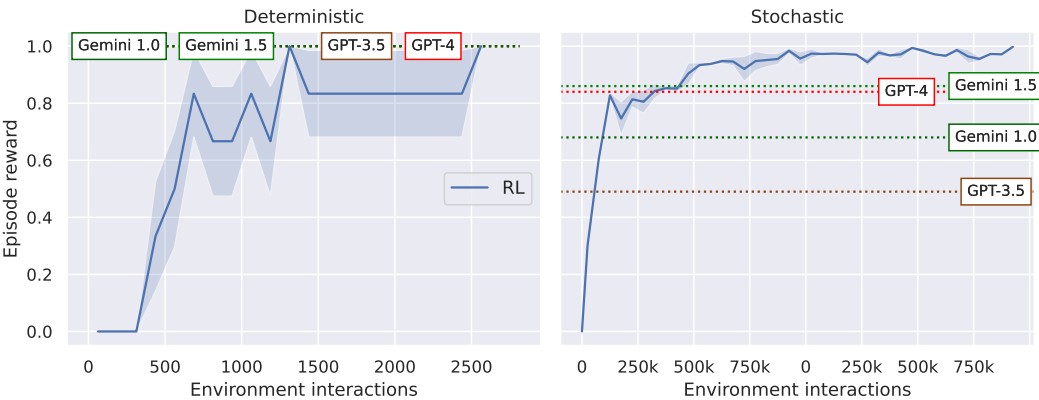

Figure 4: Decision-making results for the deterministic (left) and stochastic (right) setting. The learning curves are visible for the reinforcement learning agents learning from scratch (RL). For each foundation model, the performance of the best prompt approach is reported as a horizontal bar.

**Gridworlds.** In the first setting, we consider a $5 \times 5$ deterministic Gridworld where the reward location is fixed and known. The decision-making agent's observation consists of the current location of the agent in coordinates, as well as the location of the reward in coordinates. In the second setting, the Gridworld becomes stochastic because the reward location is now uniformly sampled. Furthermore, the agent will not have access to the location of the reward, and only observes its own location in coordinates. The latter setting poses a complex challenge, as it requires the agent to remember previous locations it visited to enable a systematic grid search.

**Algorithms.** To achieve the best possible performance, we designed and evaluated three different prompt templates that include the description of the environment, the objective, and the full history of previously visited locations but differ in the level of guidance to utilize this information:

- **Action Only (AO)**: The agent is asked to respond with an action given the current state and information available.
- **Simple Plan (SP)**: The agent is encouraged to first reason about what it should do next given the current state and information available, then decide on an action based on that plan.
- **Focused Plan (FP)**: The agent is explicitly told to use its memory to determine what position it wants to go next, then analyze which action is best to efficiently reach that position, and finally respond with that action.

These three prompts aim to encourage the agent to use different levels of reasoning and planning strategies, helping to assess how effectively LLMs can navigate decision-making tasks and adjust their behaviour based on varying levels of guidance and complexity. Particularly in the stochastic setting used in this paper, the agent should utilize its memory to know what locations it previously visited to consistently find the reward. Here, we report the best-performing strategy of each foundation model. The prompt templates, descriptions and results of all prompt-model combinations can be found in Appendix B, where each combination is evaluated for 100 independent episodes with a temperature of 0. For the reinforcement learning agents, we make use of DQN (Mnih, 2013) for the deterministic setting and Proximal Policy Optimization (Schulman et al., 2017) with a recurrent neural network (RecurrentPPO) for the stochastic and partially observable Gridworld. The latter uses a recurrent neural network to capture the temporal information necessary to explore a partially observable environment where the reward location is non-stationary. These agents interact for 1500 and 1e6 steps for the deterministic and stochastic setting, respectively. The agents are evaluated every 125 environment steps with a single episode (with a random reward location in the stochastic setting) across five random seeds.

**Results.** The results for both settings can be viewed in Figure 4. We observe that each of the foundation model agents are able to comfortably find the fixed reward location. In contrast, the

reinforcement learning agent needs several hundred environment interactions to locate the reward and learn the optimal policy, as they are not able to directly relate the reward location in the observation with the objective of the problem. In the stochastic setting, we observe that the reinforcement learning agent can learn the optimal policy, whereas none of the foundation models can yield a policy that consistently searches the entire grid.

### 3.3 CONCLUSION

The results observed in Section 3.1 suggest that foundation models' vast prior knowledge and reasoning capabilities offer promising potential for balancing exploration and exploitation in traditional reinforcement learning problems. Section 3.2 highlighted both approaches' potentially complementary strengths and weaknesses. Reinforcement learning agents often explore inefficiently as they cannot perform high-level reasoning but can learn complex optimal policies with long-term dependencies. Foundation models struggle to produce such optimal policies and, when using LLMs, are limited to text-based environments that require effortful prompt engineering. However, they can provide zero-shot efficient exploration behaviour using their prior knowledge and reasoning abilities. These observations highlight the need for a hybrid approach that utilizes the complementary strengths of both methodologies.

## 4 FOUNDATION MODEL EXPLORATION

In this section, we introduce a novel exploration scheme that combines foundation models with reinforcement learning agents. Section 4.1 discusses the observations of Section 3 to arrive at the proposed approach and describe it in detail. Section 4.2 and Section 4.3 then evaluate the efficacy of the proposed method in Minigrid (Chevalier-Boisvert et al., 2023) and Atari (Bellemare et al., 2013), respectively.

### 4.1 ALGORITHM

Building on the observations of the previous section, we propose *Foundation Model Exploration (FME)*, a hybrid exploration approach for reinforcement learning agents that incorporates exploration guidance from foundation models. FME is a general framework where a foundation model $f$, such as a language or vision-language model, controls the agent's behavior for a certain length of time $H$ instead of following the agent's original exploration policy $\pi$.

In our implementation, we introduce a multi-step exploration scheme where, with a probability $\epsilon$, the foundation model takes over action selection for the next $H$ timesteps (see Algorithm 1). We consider multi-step exploration because it enables the agent to perform coherent sequences of actions that can reach states unlikely to be visited under random or shortsighted exploration strategies. This is motivated by the challenge of deep exploration, where the agent needs to explore actions that have long-term consequences (Osband et al., 2016; 2019; Sasso et al., 2023). By leveraging the foundation model's prior knowledge and reasoning abilities, the agent can explore the environment in a more informed and directed manner. Furthermore, by

---

**Algorithm 1** FME

1: $\mathcal{D} \leftarrow$ empty (FIFO) buffer with capacity $C$
2: $s_0 \leftarrow$ initial state
3: $t \leftarrow 0$
4: **while** $t < T$ **do**
5:     $j \sim U(0, 1)$.
6:     **if** $j < \epsilon$ **then**
7:         **for** each $\tau \in \{0, \ldots, H - 1\}$ **do**
8:             **if** $t \geq T$ **then**
9:                 **break**
10:             **end if**
11:             $a_t \leftarrow f(s_t)$
12:             $r_t, s_{t+1}, \delta \leftarrow$ outcome of action $a_t$
13:             $\mathcal{D} \leftarrow \mathcal{D} \cup \{(s_t, a_t, r_t, s_{t+1}, \delta)\}$
14:             **if** $\delta = 1$ **then**
15:                 $s_{t+1} \leftarrow$ initial state
16:             **end if**
17:             $t \leftarrow t + 1$
18:         **end for**
19:     **else**
20:         $a_t \leftarrow \pi(s_t)$
21:         $r_t, s_{t+1}, \delta \leftarrow$ outcome of action $a_t$
22:         $\mathcal{D} \leftarrow \mathcal{D} \cup \{(s_t, a_t, r_t, s_{t+1}, \delta)\}$
23:         **if** $\delta = 1$ **then**
24:             $s_{t+1} \leftarrow$ initial state
25:         **end if**
26:         $t \leftarrow t + 1$
27:     **end if**
28: **end while**

---

Listing 2: The prompt template used for the VLM-based FME.

```
Explain and infer what the objective is - what action would you take from
    the actions {action_space}?
```

adjusting $\epsilon$ and $H$, we can conveniently control the foundation model's influence on both exploration and computational overhead.

We focus on off-policy algorithms in this paper for simplicity, but FME can also be adapted for on-policy algorithms. Furthermore, although FME can be demonstrated with LLMs, we showcase it with VLMs in this paper. By using VLMs, we leverage prior knowledge and reasoning abilities to infer the environment's dynamics and objectives without requiring an explicit environment description, minimizing prompt engineering to only providing the action space in natural language. As shown in Listing 2, the prompt template used for the experiments consists of a single sentence and a single variable (the action space), presented to the VLM along with the image observation. Note that we ask the VLM to justify its inference and action selection, which enables us to analyze its reasoning.

## 4.2 MINIGRID

**Environment description.** To evaluate and study the efficacy of FME, we first analyze its performance in Minigrid (Chevalier-Boisvert et al., 2023). In Minigrid environments, the objective typically is for the agent to navigate towards a reward location within a sparse reward gird world. We consider two instances that allow an in-depth analysis of the exploration behaviour of agent. The first instance is MiniGrid-Empty-$5 \times 5$, where the agent navigates to the reward location in an empty $5 \times 5$ grid. The agent has access to an image of the environment and has access to the action space $\mathcal{A} = [$*turn left*, *turn right*, *move forward*$]$. In the second instance, there is a door between the agent and the reward location, as well as a key that can be used to unlock the door. In this case, $\mathcal{A} = [$*turn left*, *turn right*, *move forward*, *pickup*, *drop*, *toggle*$]$.

**Experiments.** We study the performance of a DQN agent equipped without FME, equipped with FME powered by the VLM GPT-4o-mini, and equipped with FME powered by the VLM GPT-4o. We make use of $\epsilon = 0.01$, $H = 10$, and $N = 10$. For each DQN variant, we run the Empty instance for 5000 timesteps and the DoorKey instance for 10000 timesteps for five different seeds. The results presented in Figure 5 show that FME substantially affects the exploration behaviour, allowing significantly faster acquisition of positive rewards in both environments. Furthermore, we observe a significant gap in performance between the different VLM model sizes.

**Analysis.** As seen in Figure 5, the Minigrid environments are intuitively easy to solve from a human perspective. Similarly, the foundation models were often able to infer the objective without any explicit text information and suggest effective actions. Note that although we use Gridworlds of the smallest size, FME's advantage gap will likely increase as the environment size increases. However, we also observed that the models struggled with certain ambiguities. First, the actions *turn left* and *turn right* rotate the agent in the respective orientation from the agent's perspective, but the VLMs occasionally interpreted these orientations from the image perspective (i.e. the inverse). Furthermore, in some cases, it was unclear to the VLMs which direction the agent was currently facing, as it would occasionally reason that the triangle's hypotenuse may indicate it. Finally, in the DoorKey instances, it was not always evident to the model that the yellow block represented a door.

## 4.3 ATARI

**Environment description.** The Atari 57 benchmark is an important test bed for reinforcement learning algorithms that enables a thorough evaluation of algorithms through the various challenges posed in the different games with complex visual observations (Bellemare et al., 2013). To study the applicability of FME beyond relatively simple Gridworlds, we apply it to two well-known Atari games: Freeway and Pong. Freeway is a sparse reward environment where the objective is to cross a road whilst avoiding cars, with $\mathcal{A} = [$*NOOP*, *UP*, *DOWN*$]$. In Pong, the agent is in a dense reward

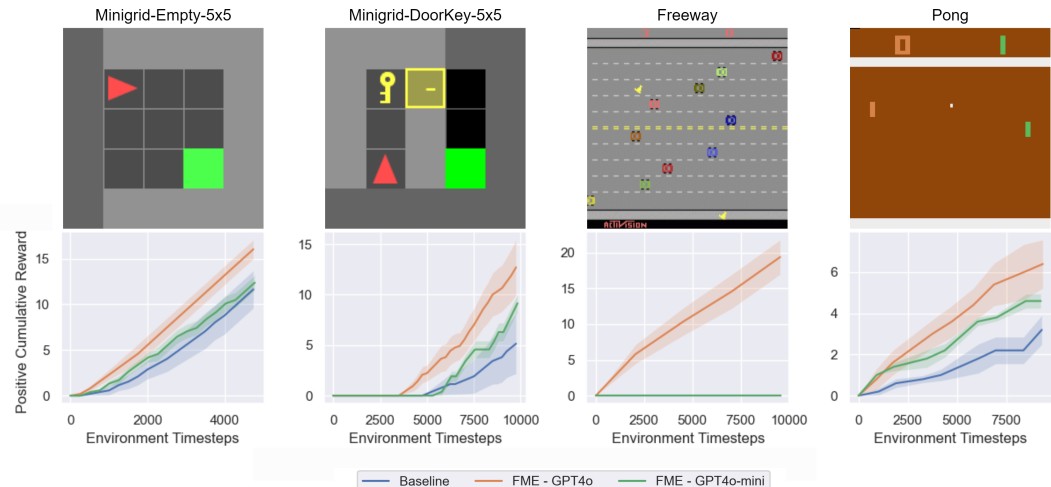

Figure 5: Performance of DQN equipped with and without FME on Minigrid-Empty-5x5, Minigrid-Doorkey-5x5, Freeway, and Pong, along with visualizations of the corresponding environments.

environment where the goal is to deflect the ball away from the agent's goal and into the opponent's goal, with action space $\mathcal{A} = [NOOP, FIRE, RIGHT, LEFT, RIGHTFIRE, LEFTFIRE]$.

**Experiments.** In this setting, we also use a DQN agent equipped without FME, equipped with FME powered by the VLM GPT-4o-mini, and equipped with FME powered by the VLM GPT-4o. To account for the increased episode durations and complexity in Atari games, we set $H = 50$. In Atari games, temporal information can be crucial to understanding the motion of objects (e.g., moving cars in Freeway or moving balls in Pong). Therefore, instead of providing a single frame, we provide the previous frame in addition to the current frame (after frameskipping) and text prompt. For each agent we study the accumulation of rewards found in the first 10000 environment interactions. The results can be observed in Figure 5. In Freeway, both the default DQN agent and the one equipped with a small VLM for FME fail to find a single reward within the time frame. In contrast, we observe that the agent equipped with the large VLM immediately starts to accumulate rewards in this sparse reward environment. In Pong, we observe that each agent is able to accumulate a positive progression of rewards but that there is a significant advantage when using FME, particularly when using the larger VLM model.

**Analysis.** As can be observed in Figure 5, the dynamics and objectives in these environments are more challenging to understand than in the Minigrid environment. We found that in the Freeway environment the GPT-4o model either recognized the game or confused it with the game 'Frogger', which has similar dynamics and objectives. This prior knowledge clearly helped with inferring the game's objective, resulting in an effective enhancement of exploration. Interestingly, the smaller VLM GPT-4o-mini rarely recognized the game nor was it able to infer its objectives, and it was even confused with other games, such as MsPacman, resulting in poor choices of actions. In Pong, we found that both VLM models were able to infer the objective and dynamics of the game. We found that there was confusion about which of the two paddles the agent controlled in some cases. However, as the models suggested actions for the paddle to which the ball was nearest, this usually did not pose a problem. Moreover, despite having access to the previous frame, we observed occasional misinterpretations of the ball direction, which could be due to a misunderstanding of the order of the provided frames.

## 5 CONCLUSION

In this paper, we explored the potential of foundation models, specifically LLMs and VLMs, to address the exploration challenges inherent in reinforcement learning. Through an in-depth analysis of their

behaviour in traditional bandit settings and Gridworld environments, we found that foundation models exhibit promising capabilities for balancing exploration and exploitation. Notably, in certain multi-armed bandit settings, GPT-4 demonstrated superior exploration efficiency compared to classical methods like Thompson Sampling and UCB, highlighting the potential of LLMs in decision-making tasks that require strategic exploration. However, our experiments in Gridworld settings revealed limitations in the ability of foundation models to reach optimal policies in more complex and stochastic environments. While LLMs could effectively navigate deterministic settings with known objectives, they struggled in environments where the reward locations were randomized and not directly observable. In contrast, the reinforcement learning agents were able to learn such complex optimal policies but would gather the required data inefficiently. These findings underscored the need for a hybrid approach that leverages the strengths of both foundation models and traditional reinforcement learning agents.

Building upon these insights, we proposed FME, an exploration approach that integrates foundation models into the reinforcement learning framework by letting a foundation model control the agent's behavior for certain periods of time. We showed that by using VLMs, FME provides intelligent exploration guidance without requiring explicit environmental descriptions or extensive prompt engineering, relying solely on the action space as a variable input. Our empirical evaluations in sparse reward Minigrid environments demonstrated that an arbitrary reinforcement learning agent can use FME in addition to a conventional exploration policy to facilitate significantly better sample efficiency. Moreover, we showed that FME scales effectively to more complex domains, such as Atari games, where it enabled agents to discover rewards more efficiently in both sparse and dense reward settings. An important observation from our experiments is that the performance of FME improves with the capacity of the underlying VLM. Larger models tend to possess more extensive prior knowledge and reasoning capabilities, which seems to translate to more effective exploration behavior. This suggests that as foundation model research progresses and more powerful models become available, the benefits of integrating foundation models into reinforcement learning for exploration purposes will likely increase.

Despite these promising results, our work also highlights several limitations and avenues for future research. One challenge is ensuring that the foundation models correctly interpret environmental cues, especially in visually complex or ambiguous settings. Misinterpretations can lead to suboptimal action choices and hinder the overall performance of the agent. Incorporating further mechanisms to provide temporal context or disambiguate visual inputs may help address these issues. Additionally, exploring alternative methods for integrating foundation models, such as different triggering mechanisms or more profound synergy between the model and the agent's policy, could further enhance exploration efficiency. Potentially interesting directions would be letting the foundation model decide autonomously if it should take control based on its confidence level or using other uncertainty measures commonly used in reinforcement learning to trigger FME. Larger scale experiments to evaluate FME, which will certainly be interesting, will also become more feasible as the cost of using foundation models decreases.

Lastly, although we did not explore providing explicit instructions or descriptions to the VLMs in this paper, it is worth considering that incorporating manual prior knowledge could further enhance the performance of foundation models in reinforcement learning tasks. Relying on such manual input reduces the generality of the solution and makes the agent dependent on specific instructions tailored to particular environments, which is why we refrained from doing so in this paper. However, by supplying additional information (which is extremely challenging with traditional reinforcement learning), researchers might enable foundation models to interpret environmental cues and objectives better, potentially overcoming some limitations observed in more complex settings in our experiments. Future research could investigate effectively integrating such information without compromising the agent's generalization capabilities.

In conclusion, our study demonstrates that foundation models have significant potential to augment exploration in reinforcement learning. By combining the high-level reasoning and prior knowledge of VLMs with the learning capabilities of reinforcement learning agents, FME offers a promising direction for developing more sample-efficient and intelligent exploration strategies. As foundation models continue to evolve, their integration into reinforcement learning frameworks could play a crucial role in overcoming the exploration-exploitation trade-off, paving the way for more efficient and effective learning in complex environments.

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

## A    MULTI-ARMED BANDIT DETAILS

**Algorithms.**    For the multi-armed bandit experiments, methods like Thompson Sampling and UCB naturally consider previous trials' outcomes by updating their belief distributions. As could be seen in Listing 1, to account for this in the decision-making for the LLMs, we provide a memory in the prompt. In this memory, we append all previous trials as '*Pulled arm {ACTION} resulting in a reward of {REWARD}*'. In the case of Thompson Sampling, we found that the best performing prior was $\alpha = 1$ and $\beta = 1$, and in the case of UCB, we used the UCB1 variant with a tuned constant $c = 0.25$.

**Prompt phrasing.**    In preliminary experiments, we tried slight variations of the prompt presented in Listing 1. For instance, we found that the LLMs significantly benefit from including "maximize the total reward by finding out which arm has the highest probability" instead of merely "maximize the total reward". We also found that providing the maximum number of trials or encouraging "an efficient exploration approach" did not have an effect on the performance.

## B    GRIDWORLD DECISION-MAKING DETAILS

**Reinforcement learning agents.**    For the reinforcement learning agents in these experiments, we used the default Stable-Baselines3 implementations of DQN and RecurrentPPO (Raffin et al., 2021). In the case of DQN, we modified the discount factor to 0.9, and in RecurrentPPO, we modified the learning rate to be 3e-5.

**Prompt templates.**    For the foundation models, the prompt templates used for the deterministic and stochastic LLM agents are almost identical. While the deterministic agent receives the exact position where the reward is located, the stochastic agent is only told the following: *Your goal is to reach the reward located at a random coordinate as quickly as possible.* See Listing 3 for the full prompt used by the deterministic LLM agent. The agent's memory is filled as it interacts with the environment. Whenever an action is executed, we add the following line to the memory: "*Executed {ACTION} at {LOCATION} resulting in {NEW LOCATION} and no reward.*" Additionally, as seen in Listing 5, whenever an agent chooses an action, it outputs a plan representing its thoughts. We add each plan to the memory as well.

Listing 3: The prompt template for the deterministic LLM agent.

```
**Context**
You are an agent in a {n}x{n} grid.
The bottom left corner is at {BOTTOM LEFT}, top left at [0, n-1], top
    right at [{n-1, n-1}], and bottom right at [{n-1}, 0].
The x-axis increases as you move rightward, and the y axis increases as
    you move upwards.
Your goal is to reach the reward located at a coordinate [{n-1},{n-1}] (
    the top-right corner).

**Memory**
[...]

**Observation**
Your current location is {OBSERVATION}

**Available Actions**
up
right
down
left

**Task**
Choose an action from the given list of actions. Output your response
    using the following JSOn format and do not use markdown.
{OUTPUT FORMAT}
```

The three prompting approaches are implemented using different JSON output formats. Note that since the action-only agent outputs no plan, its memory will only contain the results of the executed actions. Note that in the stochastic setting, we do not mention that the reward is located in the top-right corner.

Listing 4: The output format used by the **Action Only** agent.

```
{
    "action": "{The action you want to take}"
}
```

Listing 5: The output format used by the **Simple Plan** agent.

```
{
    "plan": "{Think about what you want to do next to fulfill your goal
    .}",
    "action": "{The action you want to take}"
}
```

Listing 6: The output format used by the **Focused Plan** agent.

```
{
    "plan": "{Use your memory to determine which position you want to go
    next.}",
    "analysis": "{Using your plan analyze which action is best to
    efficiently reach the position.}",
    "action": "{The action you want to take}"
}
```

**Results.** The full list of numerical results for the FA performances from the experiments in Section 3.2 can be found in Table 1, and Figure 6 and Figure 7 for the deterministic and stochastic setting, respectively. In Figure 8 example trajectories of each of the foundation models can be found in both the deterministic and stochastic settings.

Table 1: LLM Agent performances for an empty $5 \times 5$ grid with a fixed reward location and random reward location, averaged over 100 episodes.

| Model | Fixed Reward | Random Reward |
|---|---|---|
| Gemini 1.0 (AO) | 100% | 68% |
| Gemini 1.0 (SP) | 100% | 27% |
| Gemini 1.0 (FP) | 100% | 50% |
| GPT-3.5 (AO) | 100% | 43% |
| GPT-3.5 (SP) | 95% | 49% |
| GPT-3.5 (FP) | 99% | 44% |
| GPT-4 (AO) | 100% | 70% |
| GPT-4 (SP) | 99% | 84% |
| GPT-4 (FP) | 100% | 78% |
| Gemini 1.5 (AO) | 100% | 68% |
| Gemini 1.5 (SP) | 100% | 86% |
| Gemini 1.5 (FP) | 100% | 86% |

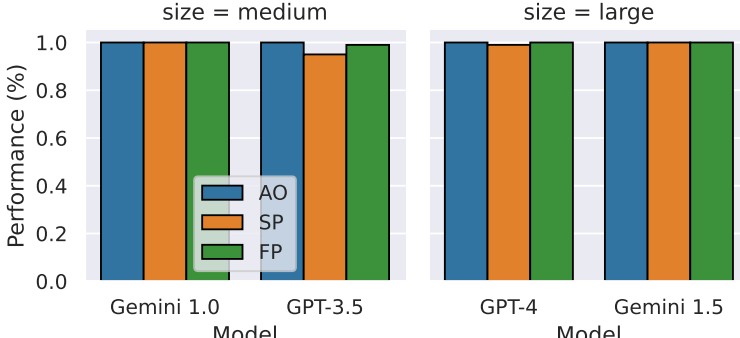

Figure 6: Performances for the FA with prompt strategies Action Only (AO), Simple Plan (SP), and Focused Plan (FP) for the deterministic setting with a fixed and known reward location.

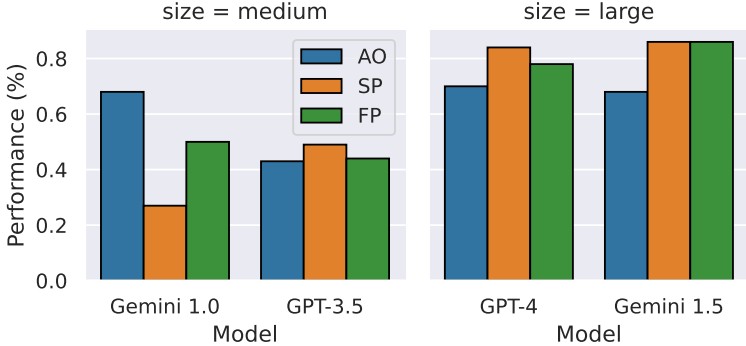

Figure 7: Performances for the FA with prompt strategies Action Only (AO), Simple Plan (SP), and Focused Plan (FP) for the stochastic setting where the reward location is randomized and unknown.

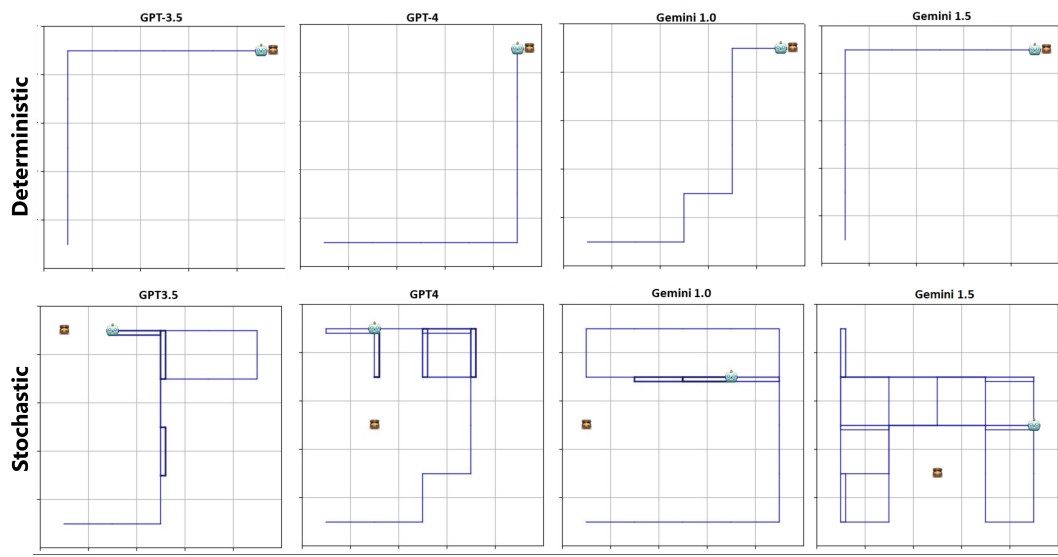

Figure 8: Example trajectories of various foundation model decision-making agents in the deterministic setting (top row) and the stochastic setting (bottom row).

## C  FME DETAILS

**Implementation.**   For our FME implementation and experiments, we used the default DQN implementation from Stable-Baselines3 (Raffin et al., 2021). For the Minigrid environments, we used a discount factor of 0.9, and for Atari, a discount factor of 0.99. Both the agent and VLM receive RGB observations from the environments. In the case of VLMs, we feed $512 \times 512 \times 3$ image observations in JPEG format with high fidelity/detail.

**Compute.**   We use OpenAI's APIs for GPT-3.5, GPT-4, GPT-4o-mini and GPT-4o, and Google Studio's APIs for Gemini 1.0 and Gemini 1.5. For training the PPO and DQN agents, we used a single NVIDIA A100 GPU in all environments. For the GPT models, we used 'GPT-3.5-turbo-0613', 'GPT-4-0613', 'GPT-4o-2024-08-06', and 'GPT-4o-mini-2024-07-18' cutoffs, which cost US$ 0,50 / 1M input tokens US$ 1,50 / 1M output tokens, US$ 30,00 / 1M input tokens US$ 60,00 / 1M output tokens, US$ 2,50 / 1M input tokens US$ 10,00 / 1M output tokens, and US$ 0,15 / 1M input tokens US$ 0,60 / 1M output tokens, respectively, as of writing. The Gemini models can be used freely as of writing, although the Gemini models have a relatively low limit for queries per minute. For the Gemini models we used 'gemini-1.0-pro-001' and 'gemini-1.5-pro'.

## D  SOCIETAL IMPACT

The methodologies proposed in this paper are primarily benign in isolation but, crucially, when future LLMs and other foundation models are leveraged as (assisting) action selectors in sequential decision-making problems for real-world applications, specific cautionary measures are necessary. For instance, the use of these models in dynamic, real-time decision-making environments such as autonomous driving would introduce significant ethical and societal challenges. To address these risks, rigorous validation processes to ensure that models behave as intended in varied and unforeseen circumstances should be used, similar to the extensive experimentation performed in this paper where we pinpoint and investigate the errors made by these models.

