# OpenReview forum: "Foundation Models for Enhanced Exploration in Reinforcement Learning"
_ICLR.cc/2025/Conference — ICLR 2025 Conference Withdrawn Submission_

### Official Review · Reviewer_b6nB · 2024-10-27

**Soundness:** 2
**Presentation:** 3
**Contribution:** 1
**Rating:** 3
**Confidence:** 5

**Summary:**

The frontier capabilities of language and vision-language foundation models are rapidly improving. The authors study the exploration behaviors of such models in simple multi-armed bandit and Gridworld environments. In the bandit setting, the performance of LLMs GPT-3, GPT-4, Gemini1.0 and Gemini1.5 is compared in the limit of trials against Thompson Sampling and UCB. In Gridworld, these same LLMs are compared zero-shot against DQN and PPO agents trained on thousands of interactions. To conclude their study, the authors introduce an exploration scheme (Foundation Model Exploration - FME) that trains RL agents by interleaving on-policy actions with (zero-shot prompted) LLM/VLM policy actions. FME is tested on DQN agents for Minigrid and the Pong and Freeway Atari games, using GPT4 and and GPT4o-mini as the teacher policies. DQN policies trained with FME vs standard exploration strategies exhibit better sample efficiency. On Freeway, FME agents exhibit better asymptotic performance.

**Strengths:**

The paper is polished and fairly well-written.

**Weaknesses:**

(1) More thorough studies of LLM exploration similar to Section 3 have been conducted in several existing works. Krishnamurthy et al. and Rahn et al. study the in-context exploration capabilities of LLMs by comparing these models against Thompson/UCB sampling in bandit tasks [1, 2]. In concurrent work, Nie et al. study this setting in even more depth, fine-tuning models on oracle demonstrations [3]. **The experiments of [1, 2, 3] suggest a conclusion that is contradictory to the fundamental premise of this paper** -- these works show that LLMs **struggle** with in-context exploration even in extremely simple bandit-like settings, unless (1) trained on demonstrations of optimal exploratory behavior or (2) prompted with CoT and summarized history. In more complex settings, works like those of Havrilla et al. [4,5] have very similar findings, demonstrating through RL experiments on LLMs that even SoTA models **do not exhibit robust exploration behaviors during decision-making**.

(2) In my opinion, the fundamental premise of FME is flawed for several reasons: (A) Querying LLMs/VLMs for actions during RL policy environment interaction amounts to sparse knowledge distillation [6, 7] -- unless the frequency of "teacher" LLM/VLM querying is decayed throughout training, the behavior of the updated RL agent may collapse to the behavior modes of the "teacher" / be upper-bounded by teacher performance; (B) The fundamental goal of RL with FME is unclear to me. If the point of FME is to try to leverage LLM/VLMs to train small policies to exhibit state-of-the-art performance in decision-making settings, why are policies trained from scratch? Why not start off by explicitly distilling knowledge to such policies by gather zero-shot demonstrations with LLM/VLMs and then continue updating these policies with RL (possibly with a KL divergence / etc or FME style additional loss term to improve the stability of training)?

(3) The models studied in Sections 3 and 4 are non-overlapping. Section 3 studies pure LLMs (Gemini, GPT-3.5, GPT-4) while Section 4 studies VLMs (GPT-4o, GPT-4o-mini). Point (1) above nonwithstanding, the authors do not confirm whether VLM models exhibit the same properties as LLM counterparts.

(4) Section 4 does not even mention what the exploration mechanism used by the DQN implementation used by the baseline policy is (epsilon greedy? something else?). Aside from the other weaknesses of the paper, this omission makes it difficult to understand what this experiment is measuring.

(5) GPT4o and GPT4o-mini are not evaluated zero-shot on the environments studied in Section 4. This omitted comparison further complicates the reader's understanding of the advantages and limitation of FME. Including it might demonstrate the extent to which the point raised in 2(B) above hold empirically; if RL agents are trained with FME *and* another exploration mechanism (e.g. epsilon-greedy sampling) *and* FME is decayed throughout training, collapse to the behavior modes of the teacher might be avoided. If true, this point would strengthen the work.

On the basis of the above points, I recommend rejection. The work has limited novelty and poor flow. Furthermore, the fundamental premise behind the FME framework -- that current LLMs/VLMs with zero-shot prompting exhibit good exploratory behaviors in complex decision-making settings -- is contradicted by existing works [1, 2, 3, 4, 5]. The utility of the method for improving the performance of models beyond SoTA RL methods and LLM/VLM prompting approaches is not demonstrated by the experiments.

[1] Krishnamurthy, Akshay, et al. "Can large language models explore in-context?." arXiv preprint arXiv:2403.15371 (2024).

[2] Rahn, Nate, Pierluca D'Oro, and Marc G. Bellemare. "Controlling Large Language Model Agents with Entropic Activation Steering." arXiv preprint arXiv:2406.00244 (2024).

[3] Nie, Allen, et al. "EVOLvE: Evaluating and Optimizing LLMs For Exploration." arXiv preprint arXiv:2410.06238 (2024).

[4] Havrilla, Alex, et al. "Glore: When, where, and how to improve llm reasoning via global and local refinements." arXiv preprint arXiv:2402.10963 (2024).

[5] Havrilla, Alex, et al. "Teaching large language models to reason with reinforcement learning." arXiv preprint arXiv:2403.04642 (2024).

[6] Kim, Yoon, and Alexander M. Rush. "Sequence-level knowledge distillation." arXiv preprint arXiv:1606.07947 (2016).

[7] Hinton, Geoffrey. "Distilling the Knowledge in a Neural Network." arXiv preprint arXiv:1503.02531 (2015).

**Questions:**

N/A

---

### Official Review · Reviewer_R6yd · 2024-10-28

**Soundness:** 2
**Presentation:** 2
**Contribution:** 2
**Rating:** 3
**Confidence:** 4

**Summary:**

This paper explores the potential of foundation models (large language models and large vision-language models) for enhancing the exploration ability of reinforcement learning agents. The authors propose Foundation Model Exploration (FME), which integrates the foundation models into the reinforcement learning framework for intelligent exploration behavior. To be specific, the authors let foundation models execute $H$ step actions with a fixed probability. The authors conduct experiments on tasks like Gridworld, and Atari to show the effectiveness of their proposed method.

**Strengths:**

## Pros

- the studied topic is interesting, i.e., exploring the possibility of foundation models in enhancing the exploration ability of the reinforcement learning algorithms
- this paper is easy to read and easy to follow, the idea is simple
- the authors provide detailed prompt information in the paper and the appendix

**Weaknesses:**

## Cons

Despite the aforementioned advantages, this paper has the following flaws:

- **Overclaim.** This paper is not the first work that utilizes VLMs to enhance the exploration efficiency of reinforcement learning agents. Some of the previous works [1, 2] have already explored that (but using different approaches). This paper only provides a heuristic method that integrates VLMs/LLMs into the exploration process of the reinforcement learning agents. Furthermore, the authors claim that they provide *the first fundamental study of the capacity of foundation models in traditional reinforcement learning exploration challenges*, but they only conduct experiments on very limited tasks and base algorithms, as well as a limited number of LLMs/VLMs. This cannot be claimed as the fundamental study.
- **The structure of this paper can be improved.** It is somewhat strange that the authors first set their focus on LLMs in bandit problems and then turn to VLMs. Section 3 and Section 4 are less connected, making it unclear what the most important insight of the paper is. If my understanding is correct, the authors would like to stress results under the VLMs, but they spend too much space on LLM results. I strongly believe that the entire experimental section in this paper can be significantly improved.
- **Lacking experiments and baselines.** This paper is also weak in that it considers too few baselines and evaluation tasks. To be specific,
  - the considered environments are too few. The authors only consider two tasks from Atari, which is not sufficient to show the effectiveness of their method. The authors should consider some hard exploration environments like Montezuma's revenge instead of some simple tasks. The authors are also encouraged to include more robotics tasks. If the authors decide to focus on the Atari benchmark, please add results on more tasks. I know it can be expensive, but they are essential to show the effectiveness of the proposed method
  - the authors should run longer environmental steps in the Pong task, as the algorithms do not converge with such limited timesteps
  - the authors include too few baseline methods. The authors should include more RL baselines (e.g., rainbow) and LLM-based methods for comparison. For example, how about directly using VLMs as the decision-maker in those environments? How about using VLMs as planners in those environments?
  - why do you only consider Gemini and GPT, how about some open-source models like Llama/Llava? It would be better to include results from other LLMs/VLMs
  - this paper lacks parameter study on the exploration probability $\epsilon$ and the VLM/LLM execution horizon $H$
- **Other issues**. There are also some other issues, including:
  - the conclusion part is too redundant. The authors should keep it concise and informative.
  - no code is provided

[1] ExploRLLM: Guiding Exploration in Reinforcement Learning with Large Language Models

[2] FuRL: Visual-Language Models as Fuzzy Rewards for Reinforcement Learning

**Questions:**

- why do you set the temperature in LLMs as 0 in bandit experiments?
- what does FIFO mean in Algorithm 1?
- what base algorithm do you use in Gridworld tasks?
- Personally, I do not think the method proposed in this paper is promising or practicable since it is extremely costly to run FME in real-world experiments. Can the authors justify the rationality and the potential future of their method? Please note that the proposed method is only evaluated to be effective in *part of the simulated tasks*. It is very unclear how it can benefit real-world tasks. I believe that directly using LLM/VLM to directly execute low-level actions is promising (i.e., leveraging the VLM/LLM itself as a decision-maker) rather than letting them output exploration actions. What do you think are the advantages of your method compared to some other methods like using LLM for constructing intrinsic rewards [1, 2], using LLM as a high-level planner [3], using LLM as a direct decision-maker [4, 5], etc.

[1] Guiding pretraining in reinforcement learning with large language models

[2] World Models with Hints of Large Language Models for Goal Achieving

[3] Large language models as commonsense knowledge for large-scale task planning

[4] Fine-Tuning Large Vision-Language Models as Decision-Making Agents via Reinforcement Learning

[5] Atari-GPT: Investigating the Capabilities of Multimodal Large Language Models as Low-Level Policies for Atari Games

---

### Official Review · Reviewer_1sQx · 2024-11-04

**Soundness:** 2
**Presentation:** 2
**Contribution:** 2
**Rating:** 3
**Confidence:** 3

**Summary:**

The paper proposes an exploration scheme for reinforcement learning (RL) in which the foundation model controls the agent for a certain period of time, dubbed Foundation Model Exploration (FME). A version of FME with a vision-language model (VLM) is studied. It is claimed that VLMs (a) reduce the amount of prompt engineering and explicit environment descriptions, (b) allow the user to specify the action space in natural language, and (c) allow the VLM to execute effective sequences of actions based on its capabilities to understand and reason about the environment's dynamics and objectives from visual input. Finally, FME is evaluated on grid worlds (Minigrid-Empty-5x5 and Minigrid-DoorKey-5x5) and two Atari games (Freeway and Pong). The paper claims that DQN equipped with FME+GPT4o outperforms its vanilla counterpart and that the scale of the gains is with the size of the underlying foundation model.

**Strengths:**

* Merging of foundation models with RL is an important line of research.
* The paper studies an important problem of exploration.
* The ability of LLMs to navigate the exploration-exploitation trade-off in a certain Bernoulli bandit is studied.

**Weaknesses:**

* The paper has some bold statements (particularly in the context of works like Voyager or Eureka), e.g., `We conduct the first fundamental study of the capacity of foundation models in traditional reinforcement learning exploration challenges,`
* Figures 1-2
	* For a specific set of bandits, a "trivial" bandit algorithm could select the optimal arm (e.g., imagine the case where the first bandit is the best).
	* In the studied case, the distribution of uniform spacings is known, and the statistics of gaps between order statistics can be computed. This may explain the results for some of the models.
	* Generally, for an unknown bandit setup, the asymptotic lower bounds for the regret are known to be logarithmic (e.g., the Lai-Robbins theorem). Consequently, it is unfair to compare general-purpose algorithms, such as UCB or Thompson Sampling, with potentially problem-specific ones.
	* It would be interesting to see some discussion concerning the possible "data leaks" about the well-known multi-armed bandit (MAB) algorithms. For instance, what do the LLMs know about general-purpose MABs, and for which bandit settings?
* In the suboptimality gap analysis, why is it claimed that $\Delta =20\%$ is a challenging scenario, considering $500$ trials?
* Experiments:
	* The underlying algorithm studied in the paper is DQN, a classical but outdated RL algorithm. There are multiple improvements upon DQN, such as Rainbow or, more recently, BBF or BRO. In particular, each of the aforementioned algorithms introduces a number of improvements on DQN, the interaction between which is unobvious. In particular, FME could help in DQN but not necessarily provide benefits for the newer line of RL algorithms.
	* The minigrid environments are toyish. What would be the result of the current approach for complex decision-making tasks like NetHack or real-life important problems, such as drug design, mathematics, etc.?
	* Setting up Atari environments includes several sharp edges, e.g., sticky actions, framestacks, non-determinism, etc. The paper should disclose the relevant choices and discuss the reasons for deviating from a classical setup if there is one (e.g., it seems that the paper uses a frame stack of 2 instead of a typical 4).
	* The experiments lack a more in-depth discussion of the results. For instance, Why do we see the plots in Figure 5? What are the main hypotheses? How could we improve the situation with additional learning (supervised or RL)? What part of DQN is improved the most? How do the results hold in more recent RL algorithms, and why? How do determinism/stochasticity or the network architecture impact the results?

**Questions:**

See above.

---

### Official Review · Reviewer_bAWg · 2024-11-04

**Soundness:** 2
**Presentation:** 3
**Contribution:** 1
**Rating:** 3
**Confidence:** 4

**Summary:**

The paper studies the application of foundation models for exploration in RL tasks. The authors first show that foundation models demonstrate good exploratory behavior in bandit tasks and simple gridworlds. Next, the authors demonstrate an approach where an RL agent defers to a foundation model action selector for parts of its trajectory. This is evaluated in sparse-reward gridworlds and Atari games. This leads to a boost in performance and the effectiveness of FME increases with the foundation models' capability.

**Strengths:**

- The paper tackles an important problem
- A good range of environments considered, from basic multi-armed bandits and gridworlds to complex Atari games
- Empirical results show gains from deferring to a foundation model for exploration
- Minimal prompt engineering which could lead to higher generalizability

**Weaknesses:**

- Missing discussion and comparison to a lot of relevant literature in exploration with foundation models, e.g. [1] studying exploration in bandit settings and [2] for minigrid BabyAI and textworld games. Given the significant overlap between the contents of this paper and related work, it is hard to identify novelty, if any, in Section 3.
- Line 87: the claim to be the first to study the capacity of foundation models to explore in RL settings is unfounded given work like [1, 2, 3]
- Insufficient justification for a hybrid approach, see [2] where foundation models can be made to solve Minigrid-like problems without any RL.
- No ablations over FME’s key hyperparameters
- Limited analysis on what kind of settings FME could succeed in and why
- No comparison to relevant exploration baselines, e.g. intrinsic reward

Minor:
- Missing citations in the first two paragraphs of the introduction
- Fig 5: no seeds information in the caption

[1] Can large language models explore in-context? Akshay Krishnamurthy, Keegan Harris, Dylan J. Foster, Cyril Zhang, Aleksandrs Slivkins.

[2] Intelligent Go-Explore: Standing on the Shoulders of Giant Foundation Models Cong Lu, Shengran Hu, Jeff Clune.

[3] LLM-Informed Multi-Armed Bandit Strategies for Non-Stationary Environments. J. de Curtò et al.

**Questions:**

- Could one more intelligently defer to the foundation model? E.g. instead of randomly, in areas with minimal progress?
- Only one RL algorithm (DQN) is considered with FME, could FME work with other RL algorithms?
- Could FME be made to work in problems with continuous action spaces?

---

### Note · Authors · 2024-11-13

I have read and agree with the venue's withdrawal policy on behalf of myself and my co-authors.